# A Review on Hydration Process and Setting Time of Limestone Calcined Clay Cement (LC3)

**Yuhan Zhao and Yingda Zhang \***

School of Architecture and Civil Engineering, Xihua University, Chengdu 610039, China
* Correspondence: yingda.zhang@mail.xhu.edu.cn

**Abstract:** The extensive usage of concrete and ordinary Portland cement has generated 5~8% of annual global $CO_2$ emissions, causing serious environmental problems. To reduce such environmental impact, researchers have made significant efforts to develop alternative materials that may partially or entirely replace the ordinary Portland cement, such as limestone calcined clay cement (LC3). LC3 has not been commonly adopted in reality because of uncertain setting times during the transportation and construction processes. Comprehensive investigation and understanding of the setting times of LC3 has great significance to industrial upgrading. As a result, this study is committed to comprehensively reviewing the hydration process and the setting time of LC3 materials.

**Keywords:** limestone calcined clay cement; hydration process; setting time; mechanical properties; durability; workability

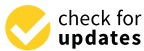



## 1. Introduction

With the fast-growing population and the continuous development of infrastructure and housing facilities, the global demands for concrete and cement have risen rapidly in the past two decades. To address today's social concerns, the cement industry has made substantial progress in improving its production efficiency and mitigating environmental impact [1].

Recent research achievements have suggested that the use of calcined clay to replace a part of limestone can significantly improve most of the cement properties. Limestone can react with alumina in the presence of calcium hydroxide, and this reaction can be intensified by adding calcined clay, which contains extra alumina. The formation of carbo-aluminate hydrates fills the pore space to enhance the mortar strength. The mixture of limestone and calcined clay with clinker is known as LC3, and its properties have been illustrated by Antonia et al. [2]. According to their findings, LC3 exhibits excellent mechanical performance when it contains only 50% clinker. As limestone is not calcined, it has only a small contribution to the emissions of $CO_2$ during the extraction and grinding process and is therefore a favorable material for the environment.

Earlier studies have shown that the manufacturing process of LC3 releases a much smaller amount of carbon dioxide than conventional cement materials and has the potential to reduce the consumption of energy and raw resources, as presented in Figure 1. Taking the cement manufacturing industry of Cuba as an example, as shown in Table 1 and Figure 2, the $CO_2$ emissions of LC3, which are the main cause of global warming, are much lower than those of other materials [3].

This paper will extensively review limestone calcined clay cement (LC3) in terms of the following three aspects: (1) history and composition of LC3 cement paste including the history of calcined clay, different types of clays, high-reactivity clay to form calcined clay, and mechanisms of metakaolin; (2) performance of LC3, such as mechanical properties, durability, and workability; (3) cement hydration, setting of Portland cement, and the effects of w/c and various materials (e.g., limestone, calcined clay, and gypsum) on hydration and setting of LC3.

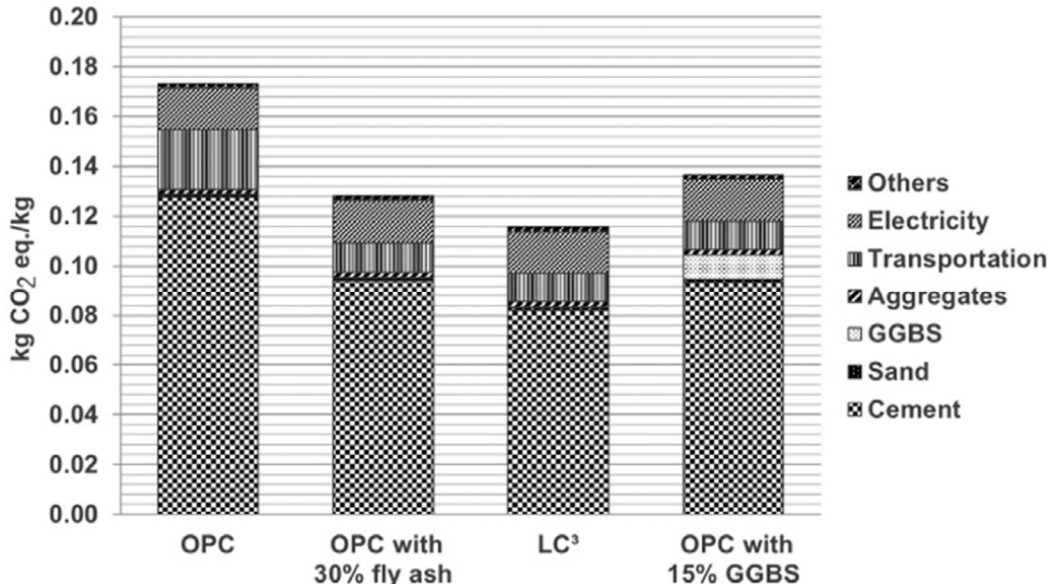

**Figure 1.** CO$_2$ emissions of LC3 in comparison with other cement. Reprinted with permission from Ref. [3]. Copyright: 2018, Karen Scrivener. Published by Elsevier.

**Table 1.** Input data of different technologies in Cuban cement industry [3]. Reprinted with permission from Ref. [3]. Copyright: 2018, Karen Scrivener. Published by Elsevier.

| Indicators | Pilot Level | Industrial Level | BAT Level |
|---|---|---|---|
| Kaolinite clay distance (km) | 150 | 60–150 | <100 |
| Type of fuel | Cuban crude oil | Pet-coke + Cuban crude oil | Gas + waste |
| Clinker technology | Wet rotary kiln | Four-stage pre-heater + pre-calciner | Six-stage pre-heater + pre-calciner |
| Clay calcining technology | Wet rotary kiln | Retrofitted calciner | Optimized flash calciner |

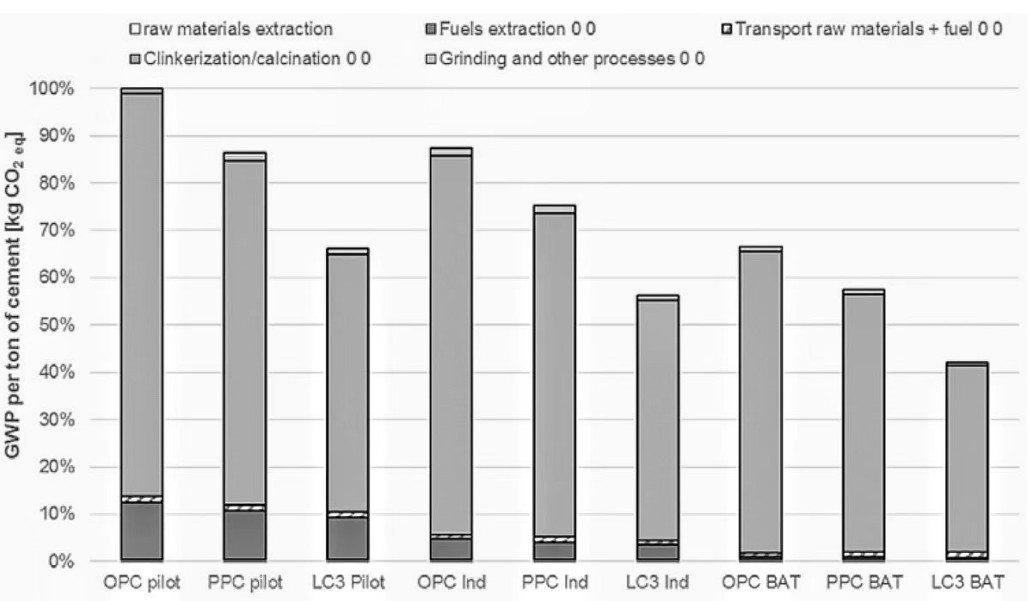

**Figure 2.** Potential impact of cement production on global warming in Cuba. Reprinted with permission from Ref. [3]. Copyright: 2018, Karen Scrivener. Published by Elsevier.

## 2. History and Composition of LC3 Cement Paste

Limestone calcined clay cement (LC3) was first proposed by the Ecole Polytechnique de Lausanne (EPFL). With the purpose of decreasing carbon emissions and optimizing resource efficiency in cement production, EPFL collaborated with partners in Cuba and India and developed LC3 as a new type of cement. In 2014, the Swiss Agency for Development and Cooperation (SADC) initiated a fund of $4 m to support this project. During the implementation of this project, researchers investigated several specific thematic aspects of cement, such as pore structure, hydrate assemblages, reactivity, rheology, durability, as well as mechanical and environmental effects. More recently, a pilot production has been conducted in India for a brand-new type of ternary blend cement, which contains 5% gypsum, 15% crushed limestone, 30% calcined clay, and 50% clinker. Out of these materials, calcined clay and limestone have been widely used as SCMs all over the world [4].

### 2.1. History of Calcined Clay

There has been a long history of using calcined clay to replace cement. As early as in the 1950s, Mielenz et al. [5,6] examined the effect of calcination on pozzolana and analyzed its potential applications in concrete production. During the 1960s, the construction of the Jupia dam of Brazil utilized metakaolin as the auxiliary cementitious material for the first time (30% of cement was replaced by SCM) [7]. Around the same period, it was reported that the Ramachandran of India also utilized finely ground calcined clays in the construction of the Bhakra Dam. Although calcined clay has long been used in industrial applications as a substitute for cement, the overall applications of clay are far less extensive than slag, fly ash, and some other SCMs. Clays are widely found all around the world in the earth crust. In the US alone, it is estimated that the amount of total reserve of kaolin exceeds 1 billion tons. Therefore, the promotion of calcined clay as a cementitious material will be conducive to the reduction in the overall construction costs in the global context [4].

### 2.2. Different Types of Clays

The weathering of silicate minerals in a variety of rocks is the primary source of clay minerals. Clay materials can be simply defined as the phyllosilicates having a particle size of about or less than 2 μm. However, it is noteworthy that some types of clays may have a much larger particle size, and some non-clay minerals can also have a particle size below 2 mm. The differences between clay and non-clay materials are mainly reflected in the chemical composition and crystal structure, which largely determine the physical properties of a material (e.g., plasticity) [4]. A single clay particle contains tens and even hundreds of layers. Its structure combines the sheets of alumina and silica. Silicon wafers are made of tetrahedral coordination of silicon atoms as shown in Figure 3. In this structure, three of the four oxygen atoms in each tetrahedron create a hexagonal net, as shown in Figure 4.

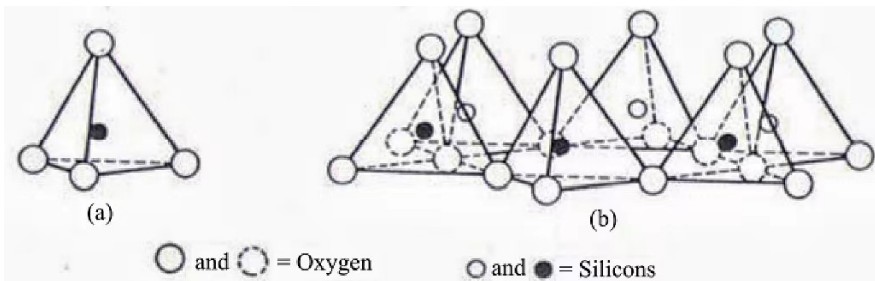

**Figure 3.** (**a**) Si location in tetrahedral; (**b**) silica sheet. Reprinted with permission from Ref. [4]. Copyright: 2013, Mathieu Antoni. Published by Swiss Federal Institute of Technology Lausanne.

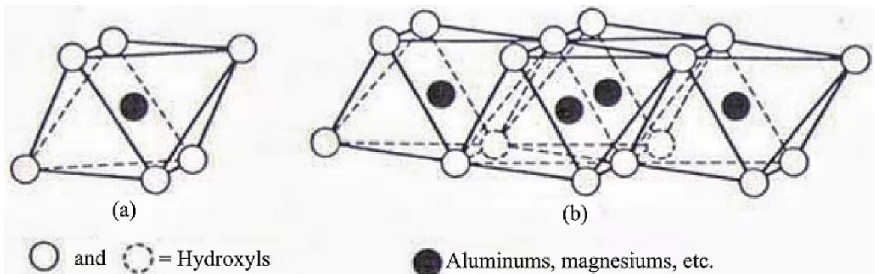

**Figure 4.** (**a**) Al location in octahedron; (**b**) alumina sheet. Reprinted with permission from Ref. [4]. Copyright: 2013, Mathieu Antoni. Published by Swiss Federal Institute of Technology Lausanne.

It is estimated that about 74% of the earth crust is made of alumina-silicate minerals, which outlines the significance and pragmatic meaning in transforming alumina-silicates into construction materials.

Clay minerals can usually be classified by the layout of arrangement and the way two- or three-sheet layers are bound together [4]. Kaolinite, illite, and montmorillonite are the three types of clays that are most widely found in the earth crust. The structures of these three types of clays are illustrated in Figure 5, and their basic characteristics are detailed in Table 2.

In most natural soils, more than one type of clay mineral can be found. Meanwhile, isomorphs substitutions commonly exist, and $Al^{3+}$ can be replaced by a variety of cations ($Fe^{2+}$, $Fe^{3+}$, $Mg^{2+}$, and $Mn^{2+}$). Furthermore, interstratification among two or more layer types may occur in a single particle, and therefore the complexity of clay structure can be extremely high. Figure 6 illustrates the diversity of the structure among different clays.

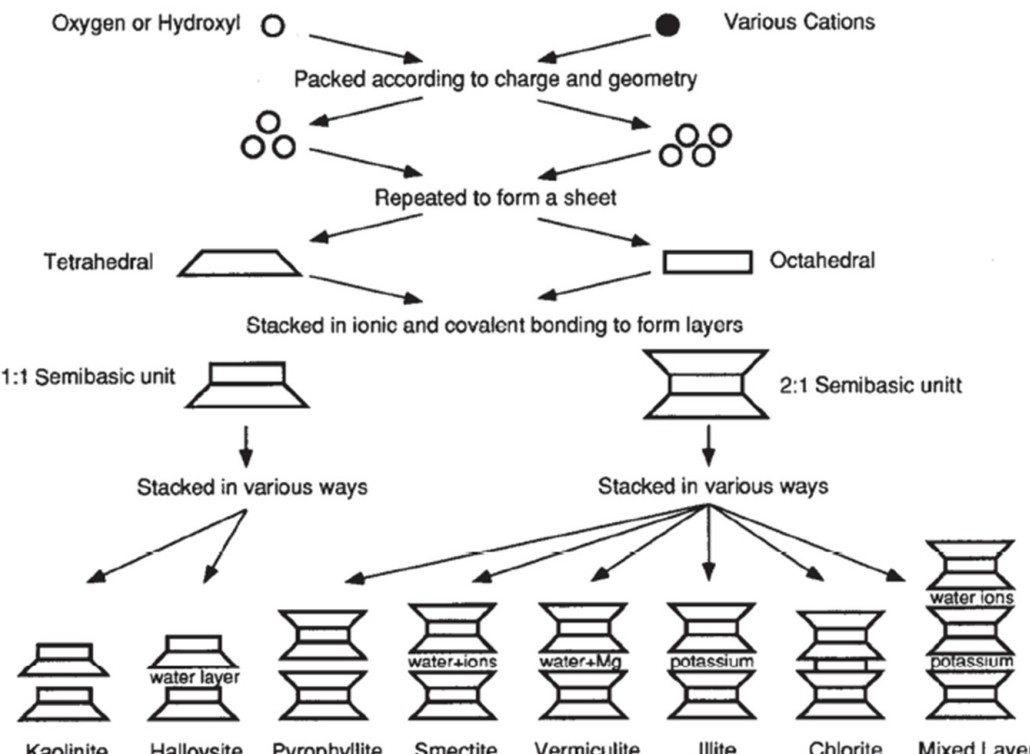

**Figure 5.** Structural charts of clay minerals. Reprinted with permission from Ref. [4]. Copyright: 2013, Mathieu Antoni. Published by Swiss Federal Institute of Technology Lausanne.

**Table 2.** Characteristics of kaolinite, illite, and montmorillonite. Reprinted with permission from Ref. [4]. Copyright: 2013, Mathieu Antoni. Published by Swiss Federal Institute of Technology Lausanne.

| Material Name | Group | Isomorphous Substitution | Interlayer Bond | Ideal Formula | Crystal System | Basal Spacing |
|---|---|---|---|---|---|---|
| Kaolinite | 1:1 | Very little | O-OH, strong | $Al_2Si_2O_5(OH)_4$ | Triclinic | 7.2 A |
| Illite | 2:1 | Some Si by Al, balanced by K between layers | K ions, strong | $(Si_4)(Al, Mg, Fe)_{2,3}O_{10}(OH)_2 \cdot (K, H_2O)$ | Monoclinc | 10 A |
| Montmorillonite | 2:1 | Mg for Al | O-O, very weak, expanding lattice | $Na_{0.33}(Al_{1.67}Mg_{0.33})Si_4O_{10}(OH)_2 \cdot n(H_2O)$ | Monoclinic | >9.6 A |

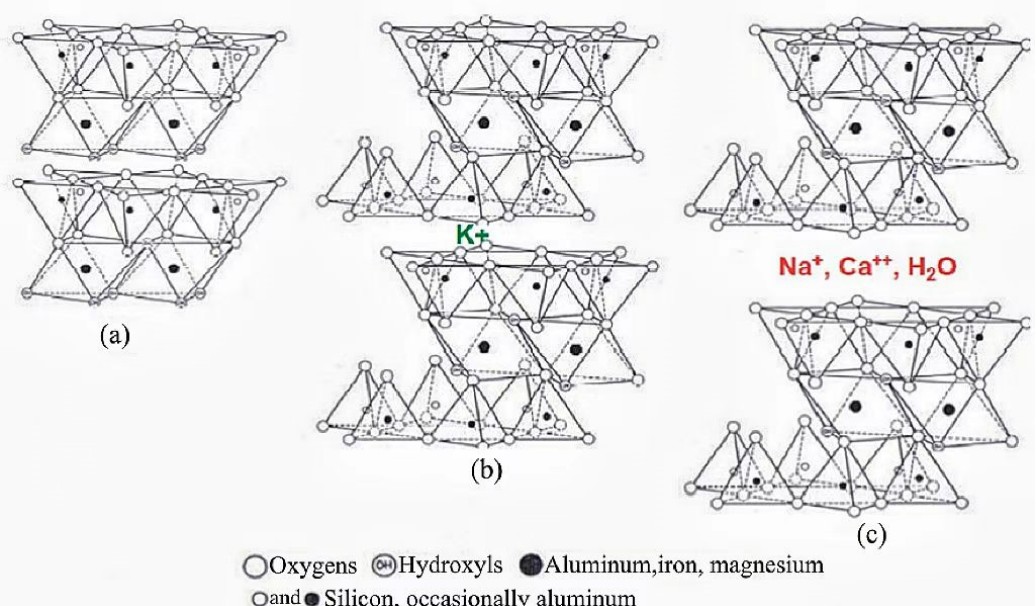

**Figure 6.** Structures of (**a**) kaolinite; (**b**) illite, and (**c**) montmorillonite. Reprinted with permission from Ref. [4]. Copyright: 2013, Mathieu Antoni. Published by Swiss Federal Institute of Technology Lausanne.

### 2.3. High-Reactivity Clay to Form Calcined Clay

He et al. [8] examined the temperature for activating different clay minerals and claimed that kaolin is characterized by the highest pozzolanic activity. Fernandez et al. [9] applied a variety of techniques to investigate the behaviors of a blend consisting of 30% calcined clay in the mortar and cement paste. Their findings showed that the blend containing metakaolin delivered the best performance. Because of the high content of aluminum in kaolinite, high-activity aluminum is generated after the calcination process, so metakaolin is characterized by high reactivity. The structural water in the clay can be removed by thermal activation at the required temperature [4]; this process is known as dihydroxylation. For kaolinite, the removal of structural water can turn clay into a metastable state with increased structural disorder, which is known as metakaolin, as shown in Figure 7.

### 2.4. Mechanism of Metakaolin

The mechanism of metakaolin in the concrete shares a high level of similarity with that of silica fume, which includes the hydration–acceleration effect, the filling compaction effect, the pozzolanic activity effect, and the inhibitory effect on alkali–aggregate reaction.

**Figure 7.** High-reactivity clays to form calcined clay. Reprinted with permission from Ref. [4]. Copyright: 2013, Mathieu Antoni. Published by Swiss Federal Institute of Technology Lausanne.

### 2.4.1. Hydration–Acceleration Effect

Metakaolin is a metastable amorphous silicon–aluminum compound. Under alkali excitation, the silicon–aluminum compound can form an aluminosilicate network structure upon depolymerization and repolymerization. After adding metakaolin into the concrete, the active $SiO_2$ and $Al_2O_3$ can rapidly react with the hydration product $Ca(OH)_2$ of the Portland cement (pozzolanic reaction) to reduce the concentration of $Ca(OH)_2$ in the liquid phase, so that cement hydration is further promoted.

### 2.4.2. Filling Compaction Effect

Concrete is a non-homogeneous body at the microstructure level, which can be deemed as a continuous grading particle accumulation system. In such a structure, the gaps among coarse aggregates are filled by fine aggregates, the gaps among fine aggregates are filled by cement particles, and the gaps among cement particles must be filled by other finer particles. The average size of cement particles is usually around 20–30 μm, while particles less than 10 μm are very few. Comparatively, the majority of metakaolin particles are below 16 μm, and the average particle size is only 10% of that of cement, so it can fill the gaps among cement particles well to increase the overall compactness of the concrete purely from the physical filling perspective. In addition, the secondary hydration of metakaolin can produce hydrated calcium silicate, hydrated calcium aluminosilicate and hydrated calcium sulfoaluminate, which all exhibit the chemical filling compaction effect. These products can further reduce the porosity and optimize the pore structure inside the concrete to help form a compact filling structure and a micro-level self-tight packing system. The combination of physical compaction and chemical filling effectively intensify the mechanical properties and durability of concrete [10].

### 2.4.3. Pozzolanic Activity Effect

During the formation of metakaolin, a large number of broken chemical bonds is generated. Therefore, metakaolin has an irregular atomic arrangement and exhibits a thermodynamically stable state with high surface energy. At the same time, the internal structure of metakaolin contains a large number of pores, for which metakaolin shows a strong pozzolanic activity. During the stirring process of concrete, the fine particles of metakaolin dissolve rapidly and absorb part of the cement hydration products $Ca(OH)_2$ (an unfavorable constituent to the strength of concrete) through secondary hydration to form C-S-H gel, $C_2AH_8$, and a small number of $C_4AH_{13}$ crystals. Consequently, the improvement of the interfacial structure between the slurry and aggregates is conducive not only to the enhancement of the mechanical properties of concrete but also to the durability of the concrete. Frías and Cabrera [11] examined the reaction between metakaolin, calcium hydroxide and water and observed that different secondary hydration products would be produced depending on the ratio of $Al_2O_3 \cdot SiO_2$ (AS$_2$) and $Ca(OH)_2$ (CH) and the reaction temperature. It should be noted that Equations (1)–(3) take place only when metakaolin

reacts with CH without limestone and gypsum. The main reaction equations include the following [12]:

$$\frac{AS_2}{CH} = 0.5, \ AS_2 + 6CH + 9H \rightarrow C_4AH_{13} + 2C - S - H \tag{1}$$

$$\frac{AS_2}{CH} = 0.6, \ AS_2 + 5CH + 3H \rightarrow C_3AH_6 + 2C - S - H \tag{2}$$

$$\frac{AS_2}{CH} = 1.0, \ AS_2 + 3CH + 6H \rightarrow C_2AH_8 + C - S - H \tag{3}$$

### 3. Performance of LC3

Three aspects related to the performance of LC3 are reviewed in this paper in comparison with other cements: mechanical properties, durability, and workability.

### 3.1. Mechanical Properties of LC3

An earlier study demonstrated that metakaolin could intensify the mechanical properties of concrete, especially at the early stage. Most of the research findings suggest that the effect of metakaolin on the strength of concrete is similar to that of silica fume. As the content of metakaolin increases, the strengths of concrete at both the early and later stage are improved [13]. The constituents of $SiO_2$ and $Al_2O_3$ in metakaolin can absorb the calcium hydroxide generated during cement hydration to form secondary C-S-H and stratlingite with gelling properties. Thus, addition of metakaolin into the cement blend can strongly intensify the early-stage strength, long-term compressive strength, flexural strength, and splitting tensile strength of concrete [14]. As reported, the effect of metakaolin on the improvement of concrete strength even surpasses that of silica fume. Most of the studies have shown that the suitable content of metakaolin in cement mortar and concrete should range from 5 to 20% by mass. Excessive addition of metakaolin may impose adverse effects on the strength of concrete, especially at the early stage. Wild et al. [15] attributed the mechanism by which metakaolin improved the strength and other properties of concrete to its filling, hydration–acceleration, and pozzolanic effects. The hydration–acceleration effect is named as the most important reason, followed by the filling effect; comparatively, the pozzolanic effect is only apparent between 7 and 14d. The research conducted by Ghafari et al. shows that addition of metakaolin can significantly reduce the amount of calcium hydroxide one day after cement hydration, so that large calcium hydroxide crystals cannot be formed at the interface between cement stones and aggregates, and thus the cement paste can gain a higher density and permeability resistance [16]. Therefore, metakaolin can be used not only to improve the strength of concrete but also to improve the durability and other properties.

It has been well established that calcined clay is an effective pozzolan for enhancing the early-stage mechanical properties and long-term strength of mortar, cement paste, and concrete. Meanwhile, Antoni's study suggested that LC3 had the same mechanical performance as Portland cement [4]. Limestone increases the total output of hydration products by improving the stabilization of ettringite and decreases the porosity in the materials due to chemical shrinkage; this is also directly related to the improvement of compressive strength [17]. Lothenbach et al. [17] claimed that addition of 5% limestone could exert a positive effect on the compressive strength of Portland cement.

Some researchers have examined the effect of combined addition of metakaolin and limestone on the various properties of cement-based materials. Because metakaolin contains a high content of aluminum phase, it can react with calcium carbonate in an alkaline environment to form calcium aluminophosphate, which may further improve the performance of the matrix. This reaction is known as the synergistic effect between metakaolin and limestone. Vance et al. [18] studied the effect of metakaolin combined with limestone of different particle sizes (average particle size: 0.7, 3, and 15 µm) on the compressive strength of cement paste. Their results indicate that, when 10% metakaolin is combined with 10%

fine limestone powder, the mechanical properties of the slurry are even better than those of 10% metakaolin alone. Antonia et al. [2] examined the change of performance of the cement mortar after it was added to a high content of metakaolin and limestone (total content of metakaolin and limestone: 15%, 30%, 45%, and 60%; mass ratio between metakaolin and limestone 2:1; water/cement ratio: 0.5). The compressive strengths of the cement paste were measured at 1, 7, 28 and 90d, respectively. Their results show that the mortar with a combined addition of 15% has a higher compressive strength than pure cement mortar at all ages. In addition, the compressive strength of the mortar with the combined addition of 30% and 45% is also higher than that of pure cement mortar at all ages except for 1d. This suggests that the proportion of cement replacement (usually less than 30%) can be further increased when metakaolin is mixed with limestone at an appropriate ratio. Alvarez et al. [19] investigated the microscopic, electrical, and mechanical properties of cement mortar mixed with metakaolin and limestone (total replacement: 15%, 25%, 30%, 40%, and 45%; content of limestone: 5%, 5%, 10%, 10%, and 15%; water/cement ratio: 0.8). Their results indicate that the 7d compressive strength of the metakaolin-limestone cement mortar is higher than that of pure cement for all mixed designs, and the 28d compressive strength is also higher than that of pure cement except for 40% replacement. Specifically, the mortar with 15% replacement has the highest compressive strength at both the early age and later age. Tironi et al. [20] examined the filling and pozzolanic effects of the Portland cement mixed with calcined clay and limestone filler (total replacement: 20% and 40%; mass ratio between calcined clay and limestone 3:1; water/cement ratio: 0.5). Their results show that the 2d compressive strength of the cement mortar with 20% replacement is higher than that of pure cement, while the 28 and 90d comprehensive strengths are close to those of pure cement. The 2 and 7d compressive strength of the cement mortar with 40% replacement is significantly lower than that of pure cement, but the strength grows rapidly at the later age and approaches that of pure cement at 90d (see Figure 8 for detailed results of compressive strength). In Figure 8, note that PC represents 100% Portland cement, 10LF represents 10% Portland cement replaced by limestone filler, 30CC represents 30% Portland cement replaced by calcined clay, 5LF15CC represents the replacement of 5% and 15% Portland cement by limestone filler and calcined clay, respectively, and 10LF30CC represents the replacement of 10% and 30% Portland cement by limestone filler and calcined clay, respectively.

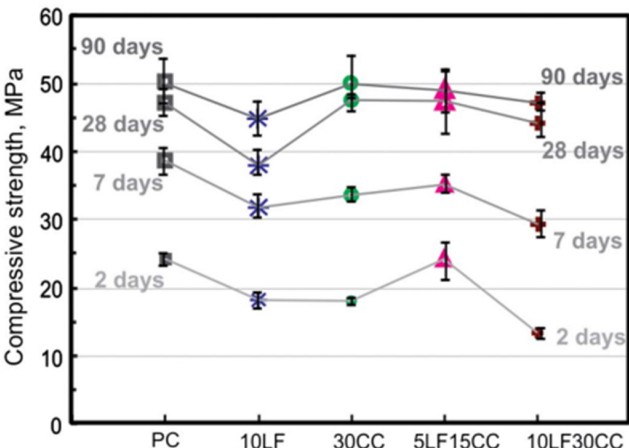

**Figure 8.** Compressive strength of mortar at different ages. Reprinted with permission from Ref. [20]. Copyright: 2017, Edgardo F. Irassar. Published by American Society of Civil Engineers.

The studies above suggest that combined mixture of metakaolin and limestone at an appropriate ratio has a positive effect on the mechanical properties of cement-based materials, mainly because: (1) at the early age, although addition of metakaolin and limestone reduces the volume and dilutes the hydration of cement, and it can also become the nucleus of cement hydration to accelerate the early-age hydration reaction; (2) at the

later stage, as metakaolin contains a high content of aluminum phase, it can react with Ca(OH)$_2$ and with limestone in an alkaline environment to form calcium mono- and hemi-aluminate, so as to further consume the hydration product Ca(OH)$_2$ and significantly improve the mechanical properties and durability of the matrix. Figure 9 shows the X-ray diffraction (XRD) patterns of the cement pastes mixed with different ratios of metakaolin and limestone at different ages [21]. The XRD pattern shows that, with combined addition of metakaolin and limestone, the diffraction peak of MC is very obvious among the various hydration products, especially for the group of 10LF30CC, which exhibits an obvious MC diffraction peak at 7d. This indicates that the synergistic reaction between metakaolin and limestone has already occurred at this point in time, and the consumption of Ca(OH)$_2$ in this group is faster than that of other groups.

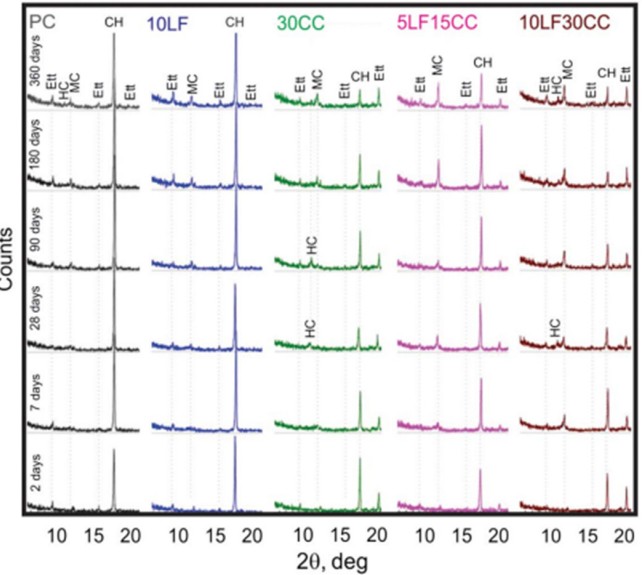

**Figure 9.** XRD pattern of the hydrated slurry at different ages. Reprinted with permission from Ref. [20]. Copyright: 2017, Edgardo F. Irassar. Published by American Society of Civil Engineers.

The aforementioned research reveals that, for the combined addition of metakaolin and limestone, the increase in metakaolin content does not significantly weaken the mechanical properties of the cement-based material. An increase in the content of metakaolin means a reduction in the volume of cement usage, which has a very positive effect on energy conservation and environmental protection. Berriel et al. [22] evaluated the environmental and economic potential of the limestone-calcined clay cement, and the results suggest that the mixture of 50% cement clinker, 15% limestone, 30% calcined clay, and 5% gypsum has a significant advantage in energy and cost savings.

### 3.2. Durability of LC3

An extensive range of studies have confirmed that the replacement of cement by an appropriate amount of metakaolin can effectively improve the permeability resistance, resistance to chloride ion permeability, resistance to corrosion, inhibitory effect on alkali–aggregate reaction, and other durability-related properties of cementitious materials [23].

In the hydration process, the active constituents of metakaolin react with Ca(OH)$_2$ (a hydration product of cement) to form more hydration products. The gelling products, which are characterized by expansive properties, can fill the pores inside the cement and the gaps between the cement and aggregates to compact the overall structure, so that the permeability resistance of the concrete can be enhanced. The chloride ion permeability is a critical indicator for the durability of marine concrete. Gruber et al. [23] examined the chloride ion permeability of concrete samples containing metakaolin substitution of 8% and 12% at w/c of 0.3 and 0.4, respectively. The results demonstrate that metakaolin can

reduce the chloride ion permeability of concrete; specifically, the samples containing 8% and 12% metakaolin at w/c = 0.4 exhibit a lower Cl diffusion coefficient than that of the reference (w/c = 0.3).

Gruber et al. [23] claimed that addition of metakaolin could significantly improve the corrosion resistance, reduce the expansive rate, and increase the strength of cement mortar; at the same time, the corrosion resistance of mortar increased with the increase in the content of metakaolin. This is largely because metakaolin admixture can reduce the content of free $Ca(OH)_2$ in the mortar. Meanwhile, $Ca(OH)_2$ exists in the vicinity of the impervious hydrated calcium silicate (C-S-H) gel, which is not conducive to the formation of swelling salts, so that the sulfate resistance of the concrete is intensified [24]. Furthermore, metakaolin can react with aluminate to reduce the formation of ettringite alongside the reaction between cement slurry and sulfate. Therefore, the sulfate solution corrosion test confirms that the concrete mixed with metakaolin has a higher strength retention rate [25].

The durability of LC3 concrete can be enhanced by using calcined clay as an SCM, as reported by Sabir et al. [25] and Siddique and Klaus [26] from the perspective of porosity reduction or water absorption reduction. Ramlochan and Thomas [21] examined the threat of sulfuric acid attack on the durability performance and concluded that sulfuric acid attack could be mitigated by adding a high content of calcined clay independent of the content of $C_3A$. In general, cement with 25% calcined clay is considered as high sulfuric acid resistant cement according to the ASTM C1012-89 standard, and the sulfuric acid resistance can be further improved by increasing the content of calcined clay.

*3.3. Workability of LC3*

Several experiments have been carried out to examine the effect of different contents of limestone and calcined clay on the overall workability of blended cement. The results generally suggest that the content of various calcined clays (e.g., kaolinite) has a significant effect on cement workability [27]. As metakaolin (another type of calcined clay) has a relatively large surface area and a high cohesion, it can significantly improve the water retention and cohesiveness of the concrete mixture and reduce the occurrence of bleeding, delamination, and segregation after being mixed with the concrete. In addition, calcined clay also has a minor effect on the fluidity of the concrete mixture. The fluidity of the concrete mixture decreases with the increase in the content of metakaolin in the mixture, but the extent of decrease is much lower than that of silica fume at the same content while like that of zeolite powder [28]. Due to a better particle shape, smaller porosity, and smaller surface area, the water demand of calcined clay is much lower than that of silica fume at the same addition ratio; therefore, the fluidity of the cement mortar with metakaolin outperforms that of cement mortar with silica fume [29].

According to the research conducted by Qian et al. [29], when the content of calcined clay is equal to 5%, its effect on the fluidity of the concrete mixture is very limited; when the content of calcined clay increases to 10–15%, the fluidity of the concrete mixture begins to decrease, but with appropriate addition of high-efficiency water reducing agent, the fluidity of the concrete mixture can be well maintained [29]. Murtaza et al. [30] conducted a comparative experiment using high-activity calcined clay and silica fume as concrete admixtures. Their results suggest that the concrete mixture with calcined clay has a lower viscosity compared to that with silica fume at the same addition ratio and the same slump. Generally, calcined clay can save 25–35% of the total usage of high-efficiency water reducing agent in comparison with silica fume, which means a significant reduction in the overall costs [30]. The results of Wild et al. show that the fluidity of the concrete mixture with both calcined clay and fly ash is higher than that of the concrete mixture with only calcined clay [31].

Earlier research has confirmed that the incorporation of calcined clay into blended cement can improve the overall performance of concrete [32]. However, it is also widely recognized that calcined clay may weaken the overall workability of concrete. Bai et al. [33] reported that the workability of the concrete containing calcined clay changed materially

at higher water/binder ratios. Lota et al. [34] claimed that the use of calcined clay as an additive rather than replacement for cement, coupled with a polymer admixture, could substantially improve the workability of the mortar.

## 4. Description of Cement Hydration and Setting

### 4.1. Portland Cement

Portland cement (PC) is usually manufactured in a rotary kiln by calcining finely ground raw materials (e.g., limestone, clay, marl, and sometimes shale) at a temperature of about 1450 °C. The product of the calcination process is named clinker and is blended with gypsum after cooling down and then ground into fine powder. The oxide constituents of the clinker include CaO (60–70%), $SiO_2$ (18–22%), $Al_2O_3$ (4–6%), and $Fe_2O_3$ (2–4%). The total of these oxide constituents accounts for about 95% of the clinker, while the remaining 5% consists of MgO, $Na_2O$, $K_2O$, $SO_3$, $Mn_2O_3$, and $TiO_2$.

Referring to Equations (4) and (5), alite and belite can both react with water to produce C-S-H gel and CH, respectively [4].

$$C_3S + zH \rightarrow C_xSHy + (3 - x)CH \tag{4}$$

$$C_2S + zH \rightarrow C_xSHy + (2 - x)CH \tag{5}$$

where $z = y + 3 - x$ for alite and $z = y + 2 - x$ for belite. The value of z typically ranges from 3 to 4. During the early stage of hydration, alite reacts with water first as the major contributing factor to the strength evolution between 7 and 28d. Comparatively, the reaction between belite and water is much slower and mainly has an effect on the strength development after 28d. The reaction between the aluminate phase, water, and gypsum produces ettringite, as shown in Equation (6). Immediately after gypsum is depleted, ettringite begins to react with the remaining aluminate to produce calcium monosulfoaluminate, as shown in Equation (7) [4].

$$C_3A + 3C\overline{S}H_2 + 26H \rightarrow C_6A\overline{S}_3H_{32} \tag{6}$$

$$2C_3A + C_6A\overline{S}_3H_{32} + 4H \rightarrow 3C_4A\overline{S}H_{12} \tag{7}$$

Similarly, the hydration of the ferrite phase shares many common ground with the hydration of the aluminate phase. Fe can replace a part of Al in ettringite and monosulfate, and the phases derived from pure ettringite and pure monosulfate with this substitution are known as alumina-ferric oxide trisulfate (AFt) and alumina-ferric oxide monosulfate (AFm). As clinker contains alkalis, which are easily soluble in the clinker phase, coupled with the formation of CH, the hydrated paste exhibits a high pH value [4].

### 4.2. Effect of Calcined Clay on Cement Hydration

Referring to Equation (8) below, the metakaolin (MK) in the pore solution of common cement is easily attacked by the hydroxyl ions and is dissolved consequently. The dissolution is enhanced by a high content of alkalinity, which may react with $Ca^{2+}$ and $OH^-$ ions and turn into C-S-H and stratlingite ($C_2ASH_8$) after precipitation.

$$Al_2O_3(SiO_2)_2 + 6OH^- + 3H_2O \rightarrow 2Al(OH)_4^- + 2H_2SiO_4^{2-} \tag{8}$$

The latter reaction in this process can be illustrated by Equation (9) as shown below:

$$AS_2 + 6CH + 6H \rightarrow C_2ASH_8 + C - S - H \tag{9}$$

With the increase in aluminum concentration in the solution, the aluminum incorporation into C-A-S-H increases accordingly, and the equation should thus be modified. Eventually, stratlingite only precipitates after the aluminum incorporation capacity of C-A-S-H is exceeded. Theoretically, some other calcium aluminate phases (e.g., $C_3AH_6$ or $C_4AH_{13}$) can precipitate as well, which has been reported for the MK-lime paste in

earlier studies [4]. However, the commonly used modern cements contain sufficiently high levels of sulfate and/or carbonate to precipitate ettringite, carboaluminates, and monosulfoaluminate. It should be also noted that hemicarboaluminate is kinetically favorable relative to monocarboaluminate; as a result, aforementioned calcium aluminate phases usually do not precipitate. The metakaolin, as a replacement of a part of cement, also has significant effects on the early-stage hydration of cement. As metakaolin can barely react with other constituents at the beginning of hydration, it exerts a dilution effect as other SCMs do, which negatively influences the strength evolution, especially in the first day. This effect appears more prominent with the increase in the substitution rate. In contrast, metakaolin may also accelerate the hydration of cement as it provides a larger surface area for heterogeneous nucleation [4].

### 4.3. Effect of Limestone on Cement Hydration

Numerous research works have been performed on the effect of limestone on hydration of cement paste. Incorporation of limestone into Portland cement was reported to not only accelerate the early-stage hydration process but also to affect the hydrate assemblage of cement paste [17]. Replacing 5–10% of cement by limestone is associated with an accelerating effect on the degree of hydration. Meanwhile, with the formation of carboaluminate phases, the pores between cement particles are filled by limestone [35]. In the hydration process of normal Portland cement, the dissolved sulfate ions can react with $C_4AF$ and/or $C_3A$ to form ettringite, and in turn, the remaining $C_3A$ and $C_4AF$ further react with ettringite to form monosulfoaluminate ($C_4A\bar{S}H_{12}$). With the existence of limestone cement (LC), carbonate ions in limestone powder interact with the aluminate hydrates generated during the hydration process to produce carboaluminates [36]. The chemical shrinkage and hydration of cement are significantly accelerated by limestone fillers from the very beginning. Such a filler effect is largely attributed to the increase in the surface area provided by limestone particles. Correspondingly, the Vicat setting time is also shortened with the addition of limestone particles. The improvement of chemical shrinkage and hydration rate largely depends on the type of cement [37], and the particle size of limestone powder influences the heat of hydration both in terms of heat rate and total heat release. It was reported that finer limestone particles (5 μm) could accelerate the hydration rate at the early stage, but the effect of larger limestone particles (20 μm) was insignificant [38].

### 4.4. Effect of Gypsum on Cement Hydration

Cement clinker condenses and releases heat immediately after stirring with water when gypsum is not added. This is mainly because the $C_3A$ in the clinker can be quickly dissolved into the water to form a condensation-accelerating calcium aluminate hydrate, which causes the cement to not work properly. Gypsum has the effect of decelerating condensation [39]. It reacts with $C_3A$ in the hydration process to produce hydrated calcium sulfate (calcium sulfate), which is barely soluble in water. The calcium sulfate precipitates on the surface of cement particles to form a protective layer, which hinders the hydration of $C_3A$ and delays the setting time of cement. If the content of gypsum is too low, the decelerating effect is insignificant. On the other hand, if the content of gypsum is too high, it can transform into a kind of condensation-enhanced substance itself, which accelerates the setting of cement. The appropriate ratio of gypsum mainly depends on the content of $C_3A$ in the cement and the content of $SO_3$ in the gypsum.

### 4.5. Effect of W/C Ratio on Cement Hydration

As the w/c decreases, both the initial setting and the final setting time of cement are reduced. The higher the w/c, the more the pores to be filled by the solid-phase hydrates, and the longer the condensation hardening process. The hardening process is largely determined by the content and speed of the generation of hydration products after cement hydration. These hydration products assemble to form an aggregate network

structure, which gradually depletes the plasticity of cement paste and eventually hardens the cement [40].

**5. Conclusions**

In recent years, the performance and durability of LC3 have attracted much attention. This paper comprehensively reviews the limestone calcined clay cement (LC3) in terms of the following three aspects: (1) history and composition of LC3 cement paste including the history of calcined clay, different types of clays, high-reactivity clay to form calcined clay, and mechanisms of metakaolin; (2) performance of LC3, such as mechanical properties, durability, and workability; (3) cement hydration, setting of Portland cement, and the effects of w/c and various materials (e.g., limestone, calcined clay, and gypsum) on hydration and setting of LC3. Through the comparative analysis of LC3, the latest research progress of LC3 is summarized and understood, and the research direction for the subsequent research on LC3 is pointed out. The major concerns are as follows:

1.  The mechanisms of hydration process of LC3 concrete needs to be further studied. The test devices such as the ultrasound pulse velocity test, Vicat apparatus, and isothermal calorimetry can be used to explore the setting time and hydration heat. The relationship between setting time and hydration heat should be clarified.
2.  The effect of mix design parameters such as w/c ratio, cement replacement, gypsum, and contents of limestone and calcined clay in LC3 cement on the setting time needs to be comprehensively considered. For instance, it is an interesting research question whether the w/c ratio or increase in the contents of limestone and calcined clay in LC3 has a positive effect on the hydration and setting time of LC3 cement.
3.  The rheological properties of LC3 cement can be further examined through various designs to develop an optimized protocol for flow curve measurement and elucidate its association with heat flow.
4.  With the development of sustainable concrete materials such as LC3 cement, LC3 concrete can be utilized in the engineering field. However, the research on LC3 cement should be extended from the hydration process and setting time to the long-term structural performance of concrete structures.

**Funding:** This research received no external funding.

**Acknowledgments:** The authors would like to thank the support from Dongting Cai.

**Conflicts of Interest:** The authors declare no conflict of interest.

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
