# Peer review of "A Review on Hydration Process and Setting Time of Limestone Calcined Clay Cement (LC3)"

_solids, doi:10.3390/solids4010003_

Round 1

Reviewer 1 Report

The authors present a work on the A Review on Hydration Process and Setting Time of Limestone 2 Calcined Clay Cement (LC3).

The article is very relevant to the wider construction industry.

In the article, the authors have carried out a very comprehensive literature review on calcined clay lime cement (LC3). The review is done in three aspects, namely:

1. the history and composition of LC3 cement paste (including the history of calcined clay, different types of clays, high-reactivity clay to form calcined clay and mechanisms of metakaolin;

2. performance of LC3 (such as mechanical properties of LC3, durability of LC3, workability of LC3)

3. illustration of cement hydration and setting for Portland cement, effect of w/c and various materials (e.g. limestone, calcined clay, gypsum) on LC3 hydration and setting.

The literature list is adequate and thematically well matched to the article. The literature list contains 40 items. Most of the literature items are from the last several years. This may indicate that the topic addressed by the authors is relevant and still current.

The reviewed article is well written in terms of content. The selection of literature and the presentation of the obtained information is done correctly. Also the cited graphs and illustrations are done correctly and clearly.

However, I have a fundamental comment on the reviewed article. The authors have limited themselves to showing only what is already published in the literature. There is no own contribution to science here. The authors have not performed any laboratory studies. The results of such studies would have been scientifically relevant, because they could have complemented (expanded) the available information on LC3. Therefore, in my opinion, the scientific value of this article in its current form is marginal. Of course, it could serve as a prelude to planning a research programme and performing research.

In conclusion, I believe that the authors should supplement the article with the results of their own research (if they have any) or at least propose directions for further work. The directions for further research should be in the conclusion and should be detailed. As it stands, the article is not suitable for publication because it does not make a significant contribution to science.

Reviewer 2 Report

This manuscript reviews the structure and property of clay minerals, and the property and hydration mechanism of LC3. Overall, the manuscript is very well written. Below are my comments that need to be addressed.

Line 33. It is not so proper to say that limestone contains no CO2 emissions, as the extraction and grinding do have CO2 emissions.

Lines 165 and 166. 1) I think it is not so proper to underline the formation of CSH-I here. Probably, you only need to say CSH. 2) C2AH8 and C4AH13 are not stable in your system due to the presence of CO32- and SO42- in LC3. Equations 1 to 3 take place only when metakaolin reacts with portlandite without limestone and gypsum.

Lines 183. Stratlingite may be better than C2ASH8.

Lines 253 and 254. Both mono- and hemi-carboaluminate can form during LC3 hydration.

Lines 288 to 298. Can you explain what do you mean by swelling salts? Do you mean ettringite here? I suggest that the authors should check if there are other theories explaining the sulfate resistance of LC3 concrete, e.g., the crystal growth theory, because the swelling theory is not so widely used in recent years.

Line 359. Hemicarboaluminate is kinetically favorable relative to monocarboaluminate

Line 396. Better to use ettringite than C4A3$H32.

Line 397. Wrong chemical formulation for monosulfate.

Round 2

Reviewer 1 Report

After the corrections introduced by the Authors, I have no comments to the article sent for review. I believe it can be published.